# Medical Needs and Therapeutic Options for Melanoma Patients Resistant to Anti-PD-1-Directed Immune Checkpoint Inhibition

**DOI:** 10.3390/cancers15133448

**Published:** 2023-06-30

**Authors:** Jessica C. Hassel, Lisa Zimmer, Thomas Sickmann, Thomas K. Eigentler, Friedegund Meier, Peter Mohr, Tobias Pukrop, Alexander Roesch, Dirk Vordermark, Christina Wendl, Ralf Gutzmer

**Affiliations:** 1Skin Cancer Center, Department of Dermatology and National Center for Tumor Diseases (NCT), University Hospital Heidelberg, 69120 Heidelberg, Germany; jessica.hassel@med.uni-heidelberg.de; 2Department of Dermatology, University Hospital Essen, 45147 Essen, Germany; lisa.zimmer@uk-essen.de (L.Z.); alexander.roesch@uk-essen.de (A.R.); 3German Cancer Consortium (DKTK), Partner Site Essen, 69120 Heidelberg, Germany; 4Bristol-Myers Squibb GmbH & Co. KGaA, 80636 Munich, Germany; thomas.sickmann@bms.com; 5Department of Dermatology, Venereology and Allergology, Charité—Universitätsmedizin Berlin, Corporate Member of Freie Universität Berlin and Humboldt-Universität zu Berlin, 10117 Berlin, Germany; thomas.eigentler@charite.de; 6Department of Dermatology, Skin Cancer Center at the University Cancer Centre and National Center for Tumor Diseases, Faculty of Medicine and University Hospital Carl Gustav Carus, Technical University Dresden, 01062 Dresden, Germany; friedegund.meier@uniklinikum-dresden.de; 7Department of Dermatology, Elbe-Kliniken, 21614 Buxtehude, Germany; peter.mohr@elbekliniken.de; 8Department of Internal Medicine III, Hematology and Oncology, University Hospital Regensburg, 93053 Regensburg, Germany; tobias.pukrop@klinik.uni-regensburg.de; 9Bavarian Cancer Research Center (BZKF), 93053 Regensburg, Germany; 10Department for Radiation Oncology, Martin-Luther University Halle-Wittenberg, 06108 Halle, Germany; dirk.vordermark@uk-halle.de; 11Department of Radiology, University Hospital Regensburg, 93053 Regensburg, Germany; christina.wendl@ukr.de; 12Department of Dermatology, Johannes Wesling Medical Center, Ruhr University Bochum, 32429 Minden, Germany

**Keywords:** melanoma, PD-1, immune checkpoint inhibition, resistance, brain metastases

## Abstract

**Simple Summary:**

Immune checkpoint blockade has dramatically improved the outcomes of patients with melanoma. Available long-term updates of clinical studies show a sustained clinical benefit for patients treated with PD-1 inhibitors such as nivolumab or pembrolizumab or for those treated with a combination of nivolumab and ipilimumab. However, about 40–50% of patients acquire resistance to therapy within five years from the start of anti-PD-1 therapy. This review assesses available definitions of the resistance and patterns of response to PD-1 immunotherapy and summarizes the potential underlying mechanisms. The available data on resistance to PD-1 therapy, medical needs and therapeutic options for melanoma patients resistant to ICI are discussed for the metastatic setting, including brain metastases, as well as for the adjuvant and neo-adjuvant settings.

**Abstract:**

Available 4- and 5-year updates for progression-free and for overall survival demonstrate a lasting clinical benefit for melanoma patients receiving anti-PD-directed immune checkpoint inhibitor therapy. However, at least one-half of the patients either do not respond to therapy or relapse early or late following the initial response to therapy. Little is known about the reasons for primary and/or secondary resistance to immunotherapy and the patterns of relapse. This review, prepared by an interdisciplinary expert panel, describes the assessment of the response and classification of resistance to PD-1 therapy, briefly summarizes the potential mechanisms of resistance, and analyzes the medical needs of and therapeutic options for melanoma patients resistant to immune checkpoint inhibitors. We appraised clinical data from trials in the metastatic, adjuvant and neo-adjuvant settings to tabulate frequencies of resistance. For these three settings, the role of predictive biomarkers for resistance is critically discussed, as well as are multimodal therapeutic options or novel immunotherapeutic approaches which may help patients overcome resistance to immune checkpoint therapy. The lack of suitable biomarkers and the currently modest outcomes of novel therapeutic regimens for overcoming resistance, most of them with a PD-1 backbone, support our recommendation to include as many patients as possible in novel or ongoing clinical trials.

## 1. Introduction

Melanoma is a life-threatening skin cancer arising from the oncogenic transformation of melanocytes with a tendency for widespread metastasis, causing around 57,000 deaths per year worldwide [1,2]. Before the development of immune checkpoint inhibition (ICI), almost all patients with metastatic melanoma died from the disease, with a median overall survival (OS) of only 6–9 months [2]. Treatment with ICI is currently the standard of care in advanced melanoma, independent of the tumor’s mutational status [3,4]. PD-1 blockade and, even more, combined PD-1 and CTLA-4 inhibition can promote durable, long-term remissions leading to 5-year OS rates of 34–44% for PD-1 inhibitors and 52% OS rates for combined ICI [5,6,7]. However, a significant proportion of advanced melanoma patients harbor or develop resistance to ICI therapy [8]. The 5-year rates of progression-free survival (PFS) range from 36% for combined nivolumab plus ipilimumab and 29% for monotherapy with the PD-1 inhibitor nivolumab to 8% for monotherapy with the CTLA-4 inhibitor ipilimumab [7]. In another landmark Phase III trial comparing the PD-1 inhibitor pembrolizumab with ipilimumab, the 4-year PFS rates for treatment-naïve melanoma patients were 27% and 8%, respectively [6]. For the newly approved ICI combination of the LAG-3 antibody relatlimab plus nivolumab, 2-year PFS data on first-line advanced melanoma again show an advantage of the ICI combination therapy over nivolumab monotherapy [9].

The efficacy of ICI in the unresectable metastatic setting led to its testing and approval in the adjuvant setting after the resection of metastasis in stage IV and III and after the resection of high-risk primary melanoma (Stage IIB/C) [10,11,12,13,14,15]. Recurrence-free survival (RFS) could be significantly improved with ICI mono- and combination therapy, however, without a proven OS benefit so far except for that of combined ICI with ipilimumab and nivolumab in stage IV disease compared to the placebo [14]. In this study, patients treated with ICI combination therapy revealed a significant RFS benefit with a HR of 0.25 (95% CI, 0.13–0.48) compared to those on the placebo and a HR of 0.41 (95% CI, 0.22–0.78) compared to those on nivolumab monotherapy. However, after 4 years, 36% (combined ICI) and 69% (anti-PD-1) of patients eventually progressed. Analyses of the CheckMate-238 study revealed a 5-year RFS rate of 50% for melanoma stage IIIB-IV patients receiving adjuvant nivolumab [13]. For patients with melanoma stage IIIA-IIID, the Keynote-054 study reported a RFS rate of 55% after 4.9 years with adjuvant pembrolizumab [15]. In stage IIB/C disease, again with adjuvant pembrolizumab, after 2 years about 19% of patients in the Keynote-716 study relapsed [10]. On the basis of these studies, pembrolizumab and nivolumab monotherapy were approved by EMA and pembrolizumab, nivolumab as well as ipilimumab monotherapy were approved by the FDA for the adjuvant treatment of melanoma.

Thus, depending on the stage of the disease, the majority of melanoma patients receiving ICI still experience refractoriness to or a relapse despite anti-PD-1 therapy at some point. This review summarizes the medical needs and therapeutic options for melanoma patients resistant to ICI.

## 2. Materials and Methods

To review the current knowledge on ICI resistance and to develop an expert opinion on the medical needs and therapeutic options for ICI-resistant patients, experts from leading German skin cancer centers were convened. A kick-off meeting was held online in November 2020 to share insights and experiences on how to advance, diagnostically and therapeutically, melanoma patients who fail anti-PD-1 therapy.. The board included physicians from dermato-oncology, oncology, radiotherapy, and radiology. Based on this input, a present narrative review was generated; the content and recommendations were agreed upon during a virtual consensus conference held in December 2022, involving the entire author panel.

The final manuscript was reviewed by the interdisciplinary panel. Standard recommendations to enhance the quality of evidence-based judgements were followed [16].

## 3. Definitions and Patterns of Response and Resistance to Immunotherapy in Melanoma

Different types of response and resistance patterns can be distinguished, as summarized in Figure 1 [17]. Response can be assessed radiologically using the classical RECIST1.1 criteria. Here, a responder (complete response (CR) or partial response (PR)) shows a decrease of at least 30% in tumor size without the occurrence of new lesions. Under ICI therapy, however, tumor shrinkage can be seen in some patients even though single new metastases occur, especially in the first weeks of treatment. Therefore, the iRECIST criteria were developed where patients with new lesions are defined as unconfirmed disease progression and can be transformed into responders without further progression in a confirmative CT scan at least 4 weeks later [18]. Moreover, an increase in size of existing lesions on radiologic images might not be based on real tumor progression but may be attributed to inflammation triggered by therapy [19] and hence, stabilize or regress later without further treatment [20]. These phenomena are called pseudoprogression. Meta-analyses reveal the overall incidence of pseudoprogression in solid tumors to be about 6% during or after ICI [21] but up to date, there are no predictive biological or clinical markers available [22].

For the classification of resistance in advanced disease, a distinction between primary resistance as non-response to therapy (i.e., disease progression (PD) or stable disease (SD) lasting less than 6 months) and secondary resistance after initial benefit (CR, PR or at least a stabilization of the disease (SD) for more than six months) is commonly made in clinical practice [23]. Due to the lack of consistent and widely accepted resistance definitions, a multi-stakeholder task force under the auspices of the Society for Immunotherapy of Cancer (SITC) has been established through consensus clinical definitions of resistance to PD-1 inhibitors in three distinct clinical scenarios: primary and secondary resistance (Figure 2), and progression after treatment discontinuation [23]. The latter has its main impact in (neo)adjuvant therapy but also for patients who have discontinued therapy due to toxicity or after having received a CR. This form of resistance could have elements of either primary or secondary resistance as defined above. The consensus considers resistances to anti-PD-1 in the metastatic, adjuvant and neo-adjuvant settings separately and incorporates relevant parameters such as a sufficient treatment duration, the pharmacodynamics of the compound (in this case anti-PD-1) and the mandatory diagnostics.

In the neo-adjuvant setting prior to resection, a clinical response according to RECIST 1.1 can be assessed via imaging [23,24]. After resection, an assessment of the pathological response is possible via a histologic review of the specimen [25]. Patients with a major pathological response (MPR, i.e., CR, near CR or major PR, ≤10% viable tumor cells) and recurrence after surgery may be classified as having secondary resistance, whereas patients without pathological response are classified as having primary resistance (Figure 2) [23]. This validation possibility is missing in the adjuvant setting, requiring a matched definition, focusing on whether or not the observed relapse occurs < or ≥12 weeks after the last dose. The 12-week criterion is based on the pharmacological assumption that anti-PD-1 inhibitors will still sufficiently occupy their receptors for this period of time after stopping therapy [26], defining an “early relapse” (primary resistance) or, conversely, a “late relapse” [23]. (Figure 2) Despite the significant progress in the consistent and comprehensive definition of anti-PD-1-related resistance provided by the SITC consensus, certain limitations and uncertainties remain, such as the value of serological markers including S100 or CT-DNA, organ specificity (local–distant metastasis; affected organs), but also the extent (single, oligo, or multiple) and dynamics of metastasis.

## 4. Potential Mechanisms of Resistance to Systemic ICI Therapy

Primary resistance reflects the “immunologic invisibility” of tumor cells. This can be due to the characteristics of the tumor cells as well as the immune system. Secondary resistance refers to an adaptive situation in which immunologic interaction leads to a selection of “immunologically invisible” tumor cells, which might occur due to the acquisition of new escape mechanisms of the selection of pre-existing immunologically invisible cells. Thus, secondary resistance is most likely due to tumor cell-intrinsic changes.

The underlying mechanisms are diverse, and similar mechanisms have been described for primary and secondary resistance (Figure 3). They can occur at the level of tumor cells (intrinsic mechanisms), e.g., defects in antigen presentation due to a low mutation load or the downregulation of MHC molecules, or alterations in interferon receptor signaling. These alterations can either result in increased type 1 interferon signaling, leading to NOS2 expression and resistance [27], or to a decrease in interferon gamma signaling [28]. Furthermore, metabolomic changes have been described in tumor cells to lead to an immunosuppressive microenvironment, such as one involving glutamine uptake and catabolism [29,30] or the exclusion of T-cells via the upregulation of beta-catenin in tumor cells [31].

Mechanisms can also occur at the level of immune cells (extrinsic mechanisms). Here, T-cells play a major role, e.g., due to the blockade of T-cell activity via hypoxia or soluble factors such as TGF beta. Moreover, the induction and recruitment of immunosuppressive cell types, e.g., regulatory T-cells (Treg) or myeloid-derived suppressor cells (MDSC) [32,33,34], as well as the presence of B-cells in tertiary lymphoid organs are important escape mechanisms [35]. These mechanisms can moreover influence each other, and usually the precise alterations are not clear in the individual case. However, there might be surrogate markers to predict the response to a checkpoint blockade or to explain secondary resistance. Among them are PD-L1 expression on the tumor as well as tumor-associated immune cells, the presence of a T-cell-inflamed tumor microenvironment signature or interferon-gamma signature [36], or a high mutational burden in the tumor cells as described in the “cancer immunogram” [37].

## 5. Resistance in the Metastatic Setting

### 5.1. Systemic Approaches

While patients with PD as the best overall response (BOR) to ICI clearly belong to the group of primary resistance, it is difficult to provide exact numbers for the frequency of acquired resistance. However, rates might be indirectly estimated from long-term PFS rates (Table 1). The estimate of the acquired resistance rate (ARr) may be deduced from PFS rates (PFSr) and PD rates (PDr); **ARr(%) = 100% − PFSr(%) − PDr(%)**. This parameter was newly developed in this review. As shown in Table 1, primary resistance occurs in approximately 30% of melanomas treated with PD-1 monotherapy and 25% of melanomas treated with the combination therapy of ipilimumab and nivolumab, whereas acquired resistance occurs in between 40% and almost 50% of melanomas under ICI therapy. The ARr depends on the depth of response in patients receiving PD-1 monotherapy; 17% of patients with CR as the best response and 54% of patients with PR as the best response eventually develop secondary resistance [7]. In patients treated with the ICI combination therapy, the risk of ARr appears to be independent of the depth of response and occurs in 20% of patients without progression within the first year after ipilimumab plus nivolumab [38]. Therefore, in patients treated with ICI combination therapy (as opposed to anti-PD-1 monotherapy), a CR does not appear to be superior to a PR in maintaining the ICI response.

Predictive biomarkers would be desirable, especially to identify primary resistant tumors and to choose upfront an alternative treatment to ICI or to delineate the reason for secondary resistance allowing rational combinations. However, as mentioned in chapter 4, no tumor-derived biomarker has been established in clinical routines yet, and few blood-derived biomarkers are used. Among them, patients with increased serum lactate dehydrogenase (LDH) levels are more likely to be primarily resistant to ICI therapy [7,43,44]. The amount of circulating tumor DNA (ctDNA) in the peripheral blood was shown to correlate with the ICI response; however, ctDNA and LDH levels might simply correlate with the tumor burden [45]. In addition, the composition of peripheral blood immune cells, especially the neutrophil–lymphocyte ratio and the number of myeloid-derived suppressor cells (MDSC) play a role [46]. High levels of soluble immune checkpoints such as sPD1 and sLAG3 have been shown to correlate with PD as BOR [47,48]. A clinically easily usable predictive biomarker would be the detection of autoantibodies; first results demonstrate that potential antibody profiles correlate with treatment response [49].

Several Phase III pivotal trials have tested various ICI combinations with varying success to decrease the number of primary resistance patients. The already approved combination of relatlimab and nivolumab within the Relativity-047 trial demonstrated an advantage of combination therapy over anti-PD-1 monotherapy, with 30% and 42% of patients revealing PD as BOR, respectively [50]. Other combinations tested, such as bempegaldesleukin plus nivolumab and the oncolytic virus talimogene laherparepvec (TVEC) plus pembrolizumab have not shown a significant benefit for patients with advanced melanoma [51,52]. A combination of PD-1 inhibition with targeted BRAF/MEK inhibition, e.g., within the Combi-I and ImSpire 150 trial, revealed a similarly low primary resistance rate as BRAF/MEK inhibition alone but marginal improvement in the development of secondary resistance [53,54,55]. However, these trials compared BRAF/MEK inhibition to BRAF/MEK inhibition plus ICI. The ImSpire 150 trial demonstrated a significant PFS improvement which led to FDA approval, whereas the COMBIi and the Keynote 22 study revealed only a trend of improved PFS without statistical significance.

Numerous Phase I–III trials for ICI-resistant patients are ongoing, with some first results reported (Table 2). The reported ORRs range between 8 and 67% with a median PFS (if reported) of 4–5 months. However, some demonstrate quite encouraging durations of response, e.g., for lifileucel, an autologous tumor-infiltrating lymphocyte (TIL) product, with the median duration of response still not reached after a median follow up of 28 months (range 2.2–35.2+ months) [56]. In the LEAP-004 study investigating the tyrosine kinase inhibitor lenvatinib plus pembrolizumab, a duration of response of 8.3 months was noted [57]. As summarized in Table 2, the studies for ICI-resistant patients are mainly Phase I and II. There is only one Phase III study, and results of this study have not been fully presented yet.

For the treatment with ipilimumab combinations after anti-PD1 failure, only a few studies have been performed. In the Illuminate studies, ipilimumab was combined with the Toll-like receptor agonist tilsotolimod, but the combination therapy was not superior to ipilimumab alone in the Phase III trial (Table 2) [70]. Concerning the comparison of ipilimumab monotherapy or combination to anti-PD-1 therapy, only retrospective analyses are available [70]. A multivariate analysis of 355 melanoma patients resistant to PD-1 (72% with primary resistance, 28% with secondary resistance) revealed that ICI combination therapy is superior to ipilimumab monotherapy in terms of response rates (odds ratio (OR) = 2.72 (95% CI = 1.5–4.93; *p* = 0.0009) and overall survival (OR = 0.61 (0.43–0.86; *p* = 0.0054), interestingly, without higher toxicity (≥grade 3 toxicity, 31% for ICI combination and 33% for ipilimumab monotherapy) [70]. A prospective Phase II study comparing ipilimumab monotherapy to a ipilimumab/nivolumab combination in PD-1 refractory patients is ongoing (SWOG S1616; NCT03033576).

Recently, ipilimumab monotherapy was compared with TIL therapy in an investigator-initiated trial with almost 170 patients. Here, PFS was 7.2 months for TIL therapy and 3.1 months for ipilimumab monotherapy (HR 0.5; 95%CI 0.35–0.72). However, overall survival was not superior and toxicity was high with all patients in the TIL arm experiencing grade 4 adverse events [71]. Another Phase II study investigated the use of TIL in the PD-1 refractory setting [71]. This approach will be further proceeded with the intent of approval by the FDA.

In BRAF V600-mutated patients, there is also the option to switch between ICI and BRAF-targeted therapy. Recent prospective sequencing studies clearly showed that patients benefit from initial ICI followed (in case of progression) by BRAF-directed targeted therapy compared to the opposite sequence [72].

### 5.2. Local Approaches

Local ablative therapies such as radiotherapy may be used, either in conjunction with systemic therapy or alone, in the case of limited disease progression in the metastatic setting. Radiation therapy induces DNA damage, autophagy and necrosis in tumor cells, leading to damage-associated molecular patterns (DAMP) and the stimulation of dendritic cells, which present tumor antigens and activate cytotoxic T-cells [73].

A recently published European consensus paper provides indication-overarching definitions for oligometastatic disease [74]. Using a system of 17 factors to characterize oligometastatic disease, the authors of the consensus paper concluded that patients with induced oligoprogression might have long-term survival when local treatment is combined with effective systemic treatment such as immunotherapy for melanoma and is used repeatedly [74,75]. However, in oligometastatic disease, the question is if a single progressing lesion develops because of secondary resistance just in that metastasis or if it is the first progressing metastasis of resistant disease. The literature is scarce on the efficacy of local ablative therapies in extracranial metastases progressing after PD-(L)1 therapy. In a retrospective study evaluating 294 patients with ICI and following solitary progression, almost half of the patients treated for solitary progression after prior response to ICI had no subsequent progression after a median follow up of 3.5 years [75]. In patients with solitary progression after the cessation of ICI, combining local therapy and ICI rechallenge was not associated with improved OS compared to the case with local treatment alone.

### 5.3. Rechallenge

Concerning rechallenge (defined as “repeated treatment with the same therapeutic class following disease progression in patients who had a clinical benefit with prior treatment for unresectable or metastatic disease”) with anti-PD-1 therapy, only retrospective analyses and small post hoc analyses of clinical trials have been published so far [76]. Within the 7-year follow up of the Phase III Keynote-006 trial, 16 patients who received a pembrolizumab rechallenge at progression after initial disease control (SD/PR/CR) were reported [77]. After rechallenge, nine patients achieved a response (56%; 4 CR, 5 PR) and five reached disease stabilization, with a 2-year PFS of 62.5%. However, from the first reports on these patients, it needs to be stated that none of the progress has been histologically verified; patients with solitary progression were additionally treated locally and 2two CRs at rechallenge were actually from adjuvant therapies [6,78]. In the largest-to-date retrospective analysis of 34 patients with metastatic melanoma who had discontinued anti-PD-1 monotherapy due to progressive disease and were rechallenged with PD-1 monotherapy, only 5 of 34 patients (14.7%) responded to retreatment [79]. This fits the results of a recent meta-analysis including seven reports on anti-PD-1 rechallenge after anti-PD-1 failure with 85 patients in total with an ORR of 15.5% and a PFS of 8.2 months [76,80]. However, it needs to be mentioned that none of the patients progressing under adjuvant anti-PD1 who continued first-line PD-1 inhibition benefitted from therapy [81]. Hence, patients might benefit from rechallenge after an interval therapy.

## 6. Resistance in Melanoma Brain Metastases

Primary and acquired resistance due to new or progressive brain metastases are common problems in metastatic melanoma patients. As registrational trials (Table 1) usually exclude patients with brain metastases or limit inclusion to those with non-active, non-symptomatic brain metastases, data for response patterns are rare, particularly for patients with symptomatic CNS metastases [82]. In recent years, a few Phase II trials have investigated the activity of ICI in asymptomatic and symptomatic melanoma brain metastases (MBM). Activated cytotoxic T-cells appear to be able to cross the blood–brain barrier, therefore providing a mechanistic rationale behind the efficacy of ICI in brain metastases [83]. In asymptomatic patients with MBM, intracranial ORR is in a similar range as that in patients with extracerebral metastases, i.e., 51–54% for nivolumab plus ipilimumab and 21%–26% for anti-PD-1 monotherapy. Intracranial PFS per a blinded, independent central review was 53% for the ICI combination after 5 years [41,42,58]. In contrast, efficacy is much lower in patients with symptomatic MBM, with an intracranial ORR of 17% and 6%, respectively; intracranial PFS was 28% after 3 years for the ICI combination therapy [41,42]. Since symptomatic MBM often requires the use of corticosteroids, this could also contribute to the lower efficacy of ICI in this patient population.

The estimated rates of intracranial primary resistance to ICI therapy remain high in MBM; they are between 30 and 40% of asymptomatic patients, and 60–70% of symptomatic patients treated with combined nivolumab and ipilimumab progress early [41,42]. For PD-1 monotherapy, the rate of PD as BOR was as high as 76% in asymptomatic patients and 80% in symptomatic patients [42,58]. Acquired resistance for patients with brain metastases can be estimated from recent data from the ABC study—18% for ipilimumab plus nivolumab and 12% for nivolumab in asymptomatic patients at 5 years (Table 1).

In tumors harboring BRAF V600 mutations, BRAF-directed therapy is also an option. In the COMBI-MB trial, response rates of intracerebral metastases to dabrafenib + trametinib were approximately 50%, and the PFS was approximately 5 months [84]. Studies are ongoing to investigate the combination of ICI- and BRAF-targeted therapy, such as the TRICOTEL study [85] and the SWOG S200 study (NCT04511013). There are also approaches combining ICI with antiangiogenic treatments in MBM, such as atezolizumab plus bevacizumab (NCT03175432).

New treatment options and combinations are therefore needed. Single-dose radiosurgery of melanoma brain metastases achieves 1-year local control rates of 86% [86,87] and comparable outcomes are achieved in larger metastases with hypofractionated radiosurgery delivered in three to five fractions [88]. Retrospective analyses suggest that the combination of stereotactic radiotherapy and immune checkpoint inhibitors achieves a survival benefit without increased toxicity compared to stereotactic radiotherapy or immune checkpoint inhibitors alone [89,90,91,92,93,94,95,96,97]. The combination strategy of immune checkpoint inhibitors plus stereotactic radiotherapy is being prospectively investigated in several studies (NCT03340129, NCT02974803 and NCT03430947). Radiosurgery might also be an option in oligoprogression with brain metastases [82]. Other interesting combination partners might include targeted therapies. Hyperactivation of the PI3K–AKT pathway, loss of PTEN expression, and activation of the oxidative phosphorylation *(OXPHOS)* metabolic pathway in melanoma brain metastases compared to metastases outside the brain might present a druggable resistance mechanism [98,99,100,101,102,103,104]. However, in PD-1-resistant patients with MBM, monotherapy with the PI3K inhibitor buparlisib did not lead to intracranial responses [105].

## 7. Resistance in the Adjuvant Setting

For ICI therapy in the adjuvant setting, the definition of resistance differs compared to the metastatic stage (Figure 2), as there is no imaging or advanced clinical method to assess response to therapy in the presence of residual (micro)metastatic disease. Therefore, the only feasible method to assess resistance is to determine the time from therapy initiation or therapy discontinuation to relapse. For adjuvant therapy, the SITC Taskforce suggested to define primary resistance/early relapse as recurrence at <12 weeks and late relapse as recurrence at ≥12 weeks after the termination of therapy (Figure 2) [23].

In six different adjuvant Phase II–III trials in patients with stage II to IV melanoma, varying recurrence rates were reported, impacted by the recurrence risk at the melanoma stage and by different follow-up times from 1–2 years in stage II to up to 4–5 years in stage III and IV (Table 3). Therapy with PD-1 inhibitors resulted in recurrence rates of between 69% (stage IV at 4 years), 45% (stage III at 5 years) and 19% (stage II at 2 years). For combined immune checkpoint blockade with a PD-1 and a CTLA-4 inhibitor, recurrence rates decreased to 36% after 4 years in stage IV melanoma; the addition of a lower dose and the use of a different schedule for ipilimumab has not shown a benefit in stage III/IV disease (Table 3) [106,107,108,109]. 

The most common type of recurrence in patients with stage III and IV disease was distant recurrence. Although this is not surprising for stage IV disease, this was also seen for stage III patients within the Keynote-054 trial, with 63% of all recurrences happening at distant sites (mainly lymph nodes, lung and liver) and 36% of all recurrences being loco-regional only at 3.5 years [111]. For stage II patients, loco-regional and distant recurrences were similarly distributed, with 4–6% at 1 year and 9% at 2 years [10,11,112]. Most frequent distant sites of metastases here were the lung, brain and lymph nodes. It is tempting to speculate that the different patterns of recurrence may relate to different mechanisms of resistance which might even be of relevance to the subsequent treatment approaches.

Concerning predictive biomarkers, the detection of ctDNA pre-treatment (with a prevalence of 16% in the adjuvant setting of stage III/IV disease) was recently shown to correlate with an increased recurrence rate within the CheckMate-915 trial (HR 1.87 for RFS; HR 2.86 for DMFS) independent of the treatment arm. An improved prediction of RFS was accomplished when the ctDNA level was combined with other biomarkers from the tumor microenvironment such as CD8+ T-cells, PD-L1, interferon-gamma (IFN-g) gene expression signature and tumor mutational burden (TMB), but also Breslow level and clinical stage [113]. In line with this, within the CheckMate-238 and Keynote-054 studies a numerical prolongation of RFS for PD-1-positive tumors was demonstrated [106,107]. In addition, an evaluation by Weber and colleagues of an IFN g gene expression signature, TMB, and CD8+ T-cell infiltration showed an association with improved RFS [114].

According to the definition by the SITC Immunotherapy Resistance Taskforce, primary resistance in adjuvant therapy could be roughly estimated from recurrence rates at 15 months (12 months of treatment + 12 weeks). Deduced from the Kaplan–Meier curves for RFS, under anti-PD-1 monotherapy, this would be about 14% of patients with stage II disease (Keynote-716), 30% of patients with stage III disease (Keynote-054) and about 54% of patients with stage IV disease (IMMUNED). In a retrospective multicenter evaluation, the median time to recurrence in stage III and IV disease patients under adjuvant anti-PD-1 therapy was 4.6 months (95% CI 0.3–35.7), where 76% experienced recurrence during adjuvant PD-1 after a median of 3.2 months and 24% experienced recurrence following treatment termination after a median of 12.5 months [81]. For patients who relapse in the adjuvant setting, further therapy selections should consider whether the relapse occurred during or after the end of adjuvant therapy, hence depending on primary or secondary resistance. Retreatment (defined as “repeated treatment with the same therapeutic class following relapse after adjuvant treatment has ended”), dose escalation (defined as treatment with the same therapeutic class) and the use of an additional agent following disease progression are possible treatment options [76,115].

Prospective data on retreatment after the failure of adjuvant ICI therapy are currently limited. Within the Keynote-054 trial, patients were retreated with pembrolizumab if the recurrence occurred more than six months after the completion of adjuvant pembrolizumab therapy. Briefly, 20 of 47 patients who developed this late relapse were retreated; of these, 9 had unresectable stage IV disease and were evaluable for response. The activity was low, with only one responder (CR 1/9; SD 3/9; PD 5/9) and a median PFS of 4.1 months [116].

In a multicenter retrospective analysis of 850 patients treated with adjuvant PD-1 inhibition in stage III or IV [81], about 17% of patients experienced recurrence within a median time of 4.6 months. Out of 136 patients with cutaneous melanoma, 104 (76%) experienced recurrence upon therapy after a median of 3.2 months, and 32 (24%) experienced recurrence after the completion of adjuvant therapy at a median time of 12.5 months. Eighty-nine (65%) patients received systemic therapy after recurrence. Of those who experienced recurrence on adjuvant treatment, none (0/6) responded to anti-PD-1 alone, 24% (8/33) responded to ipilimumab ± anti-PD-1 and 78% (18/23) responded to BRAF/MEK inhibition. Of those who recurred after adjuvant therapy, 40% (2/5) responded to anti-PD-1 retreatment in monotherapy, another 40% (2/5) responded to ipilimumab-based therapy and 90% (9/10) responded to BRAF/MEK inhibition [81]. Hence, anti-PD-1 retreatment in monotherapy might have clinical utility in patients with late relapses only. Patients with early relapses will need second-line treatments comparable to those for advanced melanoma.

In another retrospective analysis [70], out of 44 melanoma patients who relapsed after adjuvant anti-PD-1 monotherapy, 1/8 (13%) patients treated with ipilimumab monotherapy and 13/36 (36%) patients treated with ipilimumab plus PD-1 inhibitors responded to therapy, suggesting an advantage of ICI combination treatment after the failure of adjuvant anti-PD-1 [70].

Of note is that an increasing number of interventional trials are allowing the recruitment of patients with previous adjuvant therapy, often restricted to a minimum interval of recurrence-free survival after the completion of adjuvant therapy. Subgroup analyses of such patients will provide further insights into the resistance mechanism.

## 8. Lessons from the Neoadjuvant Studies

With neoadjuvant treatment, the response to systemic treatment can be assessed directly, and hence, primary resistant melanoma can easily be identified. This can be carried out radiologically before surgery as in advanced disease, but also and more precisely via a pathologic review of the surgical specimen. Interestingly, radiological and pathological responses differ a lot (Appendix A), especially concerning the rate of complete or near-complete responses. Even though a metastasis might still be seen via imaging (PR or SD according to RECIST1.1), the presence of equal to or less than 10% of viable tumor cells are per definition a major pathological response (MPR). Pooled analysis from four neoadjuvant immunotherapy trials in resectable macroscopic stage III disease revealed that patients achieving a response, and especially a MPR have a durable benefit whereas non-responders have a high risk of secondary resistance and recurrence after surgical resection [117]. The highest MPR rates were found with neoadjuvant ICI combination therapy with a 61% rate after two cycles of ipilimumab and nivolumab [117] and a 66% rate in a small study with relatlimab and nivolumab [118] as opposed to only a 21% rate with anti-PD-1 monotherapy [117] (Appendix A).

This means on the contrary that primary resistance rates are lower with ICI combination therapy with a pNR rate of 25% and 27% for nivolumab plus ipilimumab or relatlimab, respectively, and up to 66% for anti-PD-1 monotherapy [117,118].

However, in the only small trial comparing ICI combination therapy with anti-PD-1 monotherapy in the neoadjuvant setting, the difference in progression-free and overall survival was not statistically significant [119]. Importantly, patients responding to neoadjuvant immunotherapy might not need further adjuvant treatment. This was demonstrated in the PRADO study, where only patients with pathological non-response received adjuvant therapy after resection. Nevertheless, patients with a pPR not achieving a MPR had a much higher recurrence rate after surgery and might have benefitted from further adjuvant therapy to prevent secondary resistance. Hence, a pathological response might lead to further adjuvant therapy after resection. In primary melanoma, a first neoadjuvant trial with pembrolizumab was conducted on patients with desmoplastic melanoma [120], which is known to be very responsive to immunotherapy due to the high mutational burden [121]. Here, in 16 of the 29 treated patients (55%), a pCR was seen with no relapses after surgery so far.

There is some evidence that melanoma resistance can be better overcome via neoadjuvant than adjuvant immunotherapy as the stimulation of the immune response seems to result in a better expansion of tumor-resident T-cell clones in the presence of macroscopic tumors [122]. Just recently, the first randomized clinical trial was presented that compared an adjuvant with a neoadjuvant plus adjuvant pembrolizumab in more than 300 patients with resectable stage III or IV melanoma [123]. After two years, the event-free survival (event defined as not receiving surgery or adjuvant therapy, progression or death) was 72% in the perioperative and only 49% in the adjuvant treatment arm (HR, 0.58; 95% CI, 0.39–0.87). However, the OS was not statistically different between the arms. A comparison of the Kaplan–Meier curves shows that in the adjuvant arm there was especially more primary resistance based on the SITC classification criteria. This definitely supports the idea of overcoming anti-PD1 resistance with neoadjuvant therapy. Moreover, neoadjuvant therapy may also allow the escalation or de-escalation of therapy based on response; i.e., patients who have resistant disease may need escalated therapy after surgery, and vice versa. However, this should be substantiated by further randomized clinical trials, at best those including combined ICI regimens.

## 9. PD-1/PD-L1 Resistant Melanoma: Limitations, Perspectives, Conclusions

Immunotherapy is today of fundamental importance in treating melanoma, in the (neo)adjuvant as well as in the advanced setting. However, for those patients who relapse or progress during or after ICI therapy, further therapeutic options are urgently needed. The different patterns of recurrence in the adjuvant setting and of resistance in the advanced setting are presumably associated with different, diverse mechanisms of resistance to therapy with high interindividual and maybe also intraindividual differences. A better understanding of cancer resistance mechanisms is needed to develop rational strategies in clinics. However, multiple Phase I and II trials on ICI-resistant advanced/metastatic melanoma show only modest outcomes (Table 2). Most of these trials employ a PD-(L)1 backbone to demonstrate synergistic efficacy via the combined blockade of additional immunological pathways. Rational combination strategies other than PD1 plus CTLA4 or LAG3 comprise the investigation of further immune checkpoints such as TIGIT and TIM3, of epigenetic modifiers such as histone deacetylase (HDAC) inhibitors, modified cytokines (IL-2 derivatives), cytokine blockers (tocilizumab, TGFb), kinase inhibitors (lenvatinib, BRAFi+MEKi), intralesionals (TLR, oncolytic viruses), MDSC/Treg blockers, and vaccines, as well as of modifiers of tumor metabolomics. Novel biomarkers with which to characterize the immune system (e.g., microbiome, HLA, cytokines) and tumor cells/microenvironment (e.g., IFN g signature and signaling pathways) are important for an improved understanding of extrinsic and intrinsic resistance mechanisms and the rational design of therapeutic approaches [124,125,126]. Here, neoadjuvant trials might help to rapidly select the best combination therapy, at least to overcome primary resistance [127].

In the meantime, clinicians have to choose among the treatment options which are available in a type of “trial and error” approach. In addition to a switch between systemic therapies, i.e., the classical sequential use of second-, third-line and subsequent salvage therapy, these options are (i) “re-treatment”—defined as “repeated treatment with the same therapeutic class following relapse after adjuvant treatment has ended” [76,115];(ii) “rechallenge”—recently defined as “repeated treatment with the same therapeutic class following disease progression in patients who had clinical benefit with prior treatment for unresectable or metastatic disease” [76,115]; (iii) “treatment beyond progression”, i.e., treatment beyond RECIST 1.1. (or iRECIST)-defined PD [128]; or (iv) a combination with local therapies in the case of oligoprogression. However, data on the outcomes of all these approaches remain sparse and are so far deduced mainly from retrospective studies.

Therefore, the aim for all of us should be to include as many patients as possible in clinical and translational trials testing therapies to overcome anti-PD-1 resistance. Moreover, it is warranted that prospective clinical trials also continue data capturing after progression to provide evidence on the sequencing and efficacy of subsequent treatments.

## Figures and Tables

**Figure 1 cancers-15-03448-f001:**
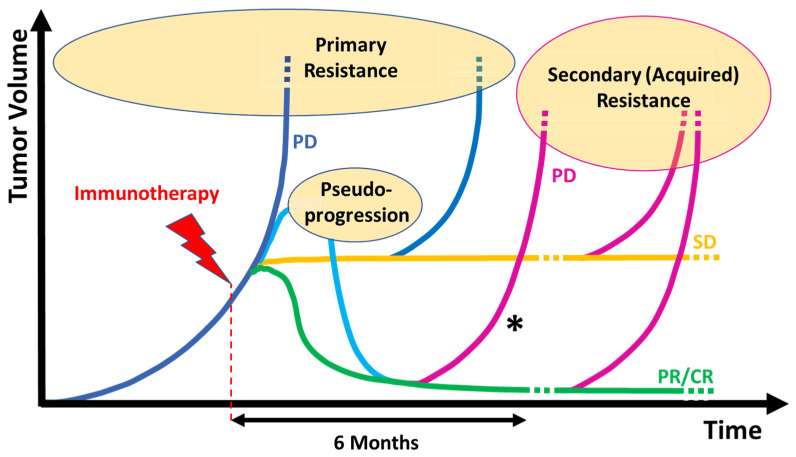
Schematic response and patterns of resistance to immunotherapy in the metastatic setting considering the SITC consensus’ clinical definition of resistance to PD-(L)1 inhibitors [23]. Primary resistance (blue line) indicates non-response to therapy including that of patients with PD or SD lasting less than 6 months. Secondary resistance (purple line) indicates resistance after initial benefit (CR, PR, and SD lasting > 6 months) and may comprise singular, oligoprogression or multiple progression emerging from a residual and/ or newly formed tumor mass. * As an extension of this rule, a progression occurring after a substantial initial benefit (PR and CR) within 6 months after the start of treatment is also considered “secondary”.

**Figure 2 cancers-15-03448-f002:**
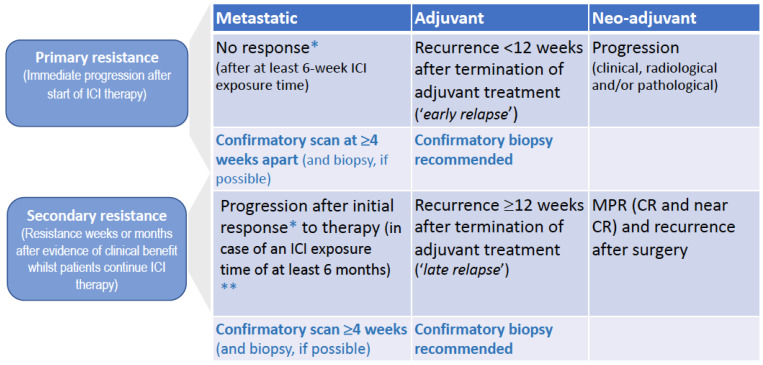
SITC scenarios and definitions of primary and secondary resistance to PD-1 therapy in the adjuvant, neo-adjuvant and metastatic treatment settings (modified to [23]). Abbreviations: ICI, immune checkpoint inhibition; MPR, major pathological response. * Complete or partial response, or stable disease for 6 months. ** As an extension of this rule, a progression occurring after a substantial benefit (PR and CR), within 6 months after the start of the treatment start is also considered “secondary”.

**Figure 3 cancers-15-03448-f003:**
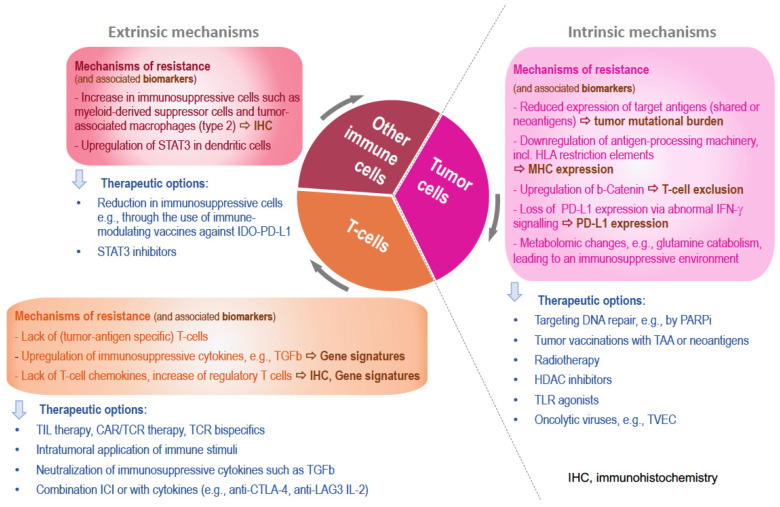
Mechanisms of resistance.

**Table 1 cancers-15-03448-t001:** Estimated frequencies of primary/acquired resistance in selected trials on advanced/metastatic melanoma (ICI monotherapy, or combined ICI). Listed are fully published data for long-term (≥4 years) follow up. To deduce acquired resistance rates, PFS rates at 4 years (48 months) and 5 years (60 months) are listed with their 95% confidence intervals (CI) (as far as reported). The rate of patients with progression (PDr) as the best overall response (BOR) serves as an estimate of the frequency of primary resistance. The estimate of the acquired resistance rate (ARr) may be deduced from PFS rates (PFSr) and PDr; **ARr(%) = 100% − PFSr(%) − PDr(%)**.

Study	Treatment Regimens	Trial Phase	Patients	N	ORR, *%* (CI)	PD Rate *%* (CI)	PFS Rate, _4 yrs_ *%* (CI)	PFS Rate, _5 yrs_ *%* (CI)	AR Rate %(For 4 yr/5 yr PFS)
*1st line ICI monotherapy (anti-PD-1)*									
Keynote-001 [39] (NCT01295827)	(**i**) Pem (2 mg/kg q3w *or* 10 mg/kg q2w *or* 10 mg/kg q3w)	IB	Adv., treatment-naïve, BRAF±	(**i**) **151**	52 (40–57)	25 (22–29)	35	29	40/44
Keynote-006 [6] (NCT01866319)	(**i**) Pem 10 mg/kg q2w vs. 10 mg/kg q3w **vs**. (**ii**) Ipi 3 mg/kg q3w x4	III	Adv., treatment-naïve, BRAF±	(**i**) **556**	42 (38–44,58–60)	29	23 (19–27)	nr	48/nr
				(**ii**) **278**	17 (12–21)	38	7 (3–13)	-	55/nr
CheckMate-066 [40] (NCT01721772)	(**i**) Nivo 3 mg/kg q2w **vs**. Dacarbazine 1000 mg/m²	III	Adv., treatment-naïve, BRAF *wild type*	(**i**) **210**	42 (36–46,58–60)	32	29	28	39/40
				(**ii**) **208**	14 (10–20)	50	3	3	47/47
CheckMate-067 [7] (NCT01844505)	(**i**) Nivo 3 mg/kg q2w (vs. Nivo + Ipi) **vs**. (**ii**) Ipi 3 mg/kg q3w x4	III	Adv., treatment-naïve, BRAF±	(**i**) **316**	45 (39–47,58–60)	38	nr	29	nr/48
				(**ii**) **315**	19 (15–24)	50	nr	8	nr/42
1st-line ICI combination therapy (anti-PD-1 + anti-CTLA-4)									
CheckMate-067 [7] (NCT01844505)	Nivo vs. (**i**) Nivo 1 mg/kg q2w + Ipi 3 mg/kg **vs**. (**ii**) Ipi 3 mg/kg q3w x4	III	Adv., treatment-naïve, BRAF±	**(i) 314**	58 (50–57,61–64)	24	nr	36	nr/40
				**(ii) 315**	19 (15–24)	50	nr	8	nr/42
*1st-line ICI therapy in patients with asymptomatic brain metastasis (anti-PD-1 ± anti-CTLA-4) for intracranial outcome parameters*									
CheckMate-204 [41] (NCT02320058)	(**i**) Nivo 1 mg/kg q2w + Ipi 3 mg/kg **^a^**	II	Adv., treatment-naïve **^b^** ABM, BRAF±	(**i**) **101**	54 (40–57,61–64)	30	nr	nr	-
ABC [42] (NCT02374242)	Nivo 1 mg/kg q2w + Ipi 3 mg/kg **vs**. Nivo 3 mg/kg	II	Adv., treatment-naïve ABM, BRAF±	**27**	59	30	nr	52	nr/18
				**19**	21	74	nr	14	nr/12

^a^ Same treatment regimen as that in CheckMate-067; ^b^ previously approved adjuvant systemic therapies were allowed, including ipilimumab, if the last dose was administered at least 6 months before the first dose of the study drug was (criterium applies to CheckMate-204 trial only). Previous use of BRAF and MEK inhibitors in the advanced setting was allowed (after a washout period; 5 patients received previous BRAF/MEK inhibitors). Abbreviations: ABM, asymptomatic brain metastases; BRAF±, patients with BRAF mutations and wild-type BRAF; CI, 95% confidence interval; Ipi, ipilimumab; mets, metastasis; Nivo, nivolumab; nr, not reported; ORR, overall response rate; Pem, pembrolizumab; PD, progressive disease; PFS, median progression-free survival; yr/yrs, year(s).

**Table 2 cancers-15-03448-t002:** Efficacy outcomes reported for Phase I–II trials in advanced/metastatic, ICI-resistant melanoma.

Trial (NCT n°)	Treatment Regimens	Trial Phase	Patients	N	Primary Endpoint	ORR, % (CI)	PFS _Median_, mts (HR (CI))	OS _Med._, mts (HR (CI))
*2nd-line combination therapy (anti-PD-1 backbone)*								
LEAP-004 [57] (NCT03776136)	Lenvatinib 20 od + Pem 200 mg q3w	II	Adv., PD-(L)1 pre-treated (PD upon/after therapy ^a^) [58], BRAF±	**103**	ORR	33 (17–53) ^b^; 23 (13–35) ^c^	4.2 (3.8–7.1)	14.0 (10.8-nr)
IRB17-0686 [59] (NCT02743819)	Ipi 1 mg/kg q3w × 4 + Pem 200 mg q3w	II	Adv., PD-(L)1 pre-treated (PD up**on** therapy ^d^ *= Primary* *resistance*), BRAF±	**70**	ORR	31 (nr-nr)	4.7 (2.8–8.3)	nr
CA224-020 [60] (NCT01968109)	Relatlimab 80 mg q2w + Nivo 240 mg q2w	I–II	Adv., PD-(L)1 pre-treated (PD up**on** ther. *= Prim. resistance*), BRAF±	**68**	Safety, ORR	12 (nr-nr)	nr	nr
SYNERGY-001 [61] (NCT02521870)	SD-101 2 mg/kg q1-3w + Pem 200 mg q3w	I–II	Advanced, PD-(L)1-pre-treated (PD upon/after ther.), BRAF±	**23**	ORR	20 (nr-nr)	nr	nr
2014-0922 [62] (NCT02500576)	Cryopreserved TIL**^e^** + IL-2 (Aldesleukin) (high dose/low dose) + Pem 200 mg q3w	I–II	Metastatic, un-/pre-treated (13 out of 14 pts were PD-1-pre-treated), BRAF±	**14**	ORR	14 (nr-nr)	3.9 (nr-nr)/2.1 (nr-nr)	9.7 (nr-nr)/ 8.8 (nr-nr)
PV-10-MM-1201 [63] (NCT02557321)	PV-10 (intralesional) + Pem 2 mg/kg q3w	I	Metastatic, ICI-pre-treated (PD upon/after therapy), BRAF±	**13**	Safety	31 (nr-nr)	nr	nr
4SC-202-2-2017 [64] (NCT03278665)	Domatinostat + Pem 2 mg/kg q3w	I–II	Metastatic, ICI-pre-treated (PD upon/after therapy), BRAF±	**40**	Safety	8 (nr-nr)	nr	nr
16-1080.cc [65] (NCT03200847)	all-*trans*-Retinoic acid + Pem 200 mg q3w	I–II	Metastatic, ICI pre-treated (PD upon/after therapy), BRAF±	**24**	Safety	67 (nr-nr)	20.3	nr
Lipo-MERIT [66,67] (NCT02410733)	FixVak (RNA vaccine) ± anti-PD-1	I	Metastatic, ICI-pre-treated (PD on/after therapy), BRAF±	**42**	Safety	16 (FV mono) 35 (FV + PD1)	nr	nr
*2nd-line combination therapy (anti-CTLA-4 backbone)*								
ILLUMINATE-204 [68] (NCT02644967)	Tilsotolimod 8 mg/kg q1-6w + Ipi 3 m/kg q3w x4	I–II	Advanced, PD-1-pre-treated (PD on/after therapy), BRAF±	**62**	Safety, ORR	22 (12–37)	5.1 (3.7–7.0)	21.0 (9.8-nr)
ILLUMINATE-301 [69] (NCT03445533)	Tilsotolimod 8 mg/kg q1-6w + Ipi 3 m/kg q3w x4 **vs**. Ipi	III	Advanced, PD-1-pre-treated	**481**	OS and ORR	9 (nr-nr)	nr	nr
*2nd-line monotherapy*								
C144-01 [56] (NCT02360579)	Lifileucel (i.e., autologous, cryo-preserved TIL **^e^**) + IL-2 x6	II	Advanced, PD-1-pre-treated, BRAF±	**66**	ORR	36 (nr-nr)	nr	nr
Dutch [56] (NCT02278887)	TIL (i.e., autologous, cryo-preserved TIL **^e^**) **vs**. Ipi	III	Advanced, progression after the maximal one line of pre-treatment (no Ipi), BRAF±; approximately 90% of patients had PD-1 pre-treatment in both arms	**84** **84**	PFS	48.8 21.4	7.2 (4.2–13.1)/3.1 (3.0–4.3), HR 0.05, *p* < 0.001	25.8 (18.2-nr) /18.9 (13.8–32.6), HR 0.83, *p* = 0.39

^a^ Upon therapy or ≤12 weeks after last dose of an anti-PD-(L)1 agent given alone or in combination (including with anti-CTLA-4 therapy) for ≥2 doses; ^b^ PD upon prior anti-PD-1 plus CTLA-4 therapy; ^c^ primary resistance to prior anti-PD-(L)1 monotherapy; ^d^ PD (or stable disease lasting ≥24 weeks during treatment with an anti-PD-(L)1 antibody as the treatment regimen) immediately prior to recruitment to this study or PD within ≤6 months of adjuvant anti-PD1 antibody administration; ^e^ harvested TIL re-administered in patients after lymphodepleting chemotherapy comprising cyclophosphamide and fludarabine phosphate. Abbreviations: BEMPEG, Bempegaldesleukin; BICR, blinded, independent central review; BRAF±, mutated BRAF and wild-type BRAF; CI, 95% confidence interval; FV, FixVak; IA, investigator-assessed; ICI, immune checkpoint inhibitor; Ipi, Ipilimumab; MDSC, myeloid-derived suppressor cells; Nivo, Nivolumab; NR, not reported (and/or: not reached); od, once daily; ORR, overall response rate; OS, median overall survival; PD, progressive disease; Pem, Pembrolizumab; PFS, median progression-free survival; PV-10, 10% rose bengal disodium for injection; SD-101, synthetic CpG-ODN agonist of TLR 9; TIL, tumor-infiltrating lymphocytes.

**Table 3 cancers-15-03448-t003:** Estimated frequencies of resistance in selected trials investigating the adjuvant use of ICI in melanoma (ICI monotherapy, or combined ICI).

Study	Treatment Regimens	Trial Phase	Patients (Stage Distribution)	N	RFS rate, *%* (CI)	Recurrence Rate, % (5 yr/*3 yr/4 yr) ^a^	Patterns of Recurrence: Local vs. (//) Distant Only % (n/n_total_) ^b^
*Stage III/IV*							
Keynote-054 [15] (NCT02362594)	(**i**) Pem 200 mg q3w vs.	III	100% III	**(i) 514**	55 at 5 yrs (51–60)	45	14 (74/514)//28 (143/514)
	(**ii**) Placebo		100% III	**(ii) 505**	38 at 5 yrs (34–43)	62	19 (96/505//41 (206/505))
CheckMate-238 [13] (NCT02388906)	(**i**) Nivo 3mg/kg q2w vs.	III	81% III, 18% IV	**(i) 453**	50 at 5 yrs (not reported)	nr 50	
	(**ii**) Ipi 10mg/kg q3w x4		81% III, 19% IV	**(ii) 453**	39 at 5 yrs (not reported)	nr 61	
CheckMate-915 [110] (NCT03068455)	(**i**) Nivo 480 mg q2w vs. (ii) Nivo	III	86% III, 13% IV	**(i) 924**	63 at 2 yrs A (60–66)	-	nr//nr
	240 mg q2w + Ipi 1 mg/kg q6w		87% III, 13% IV	**(ii) 920**	65 at 2 yrs (61–68)	-	nr//nr
IMMUNED [14] (NCT02523313)	(**i**) Nivo 3mg/kg q2w vs.	II	100% IV	**(i) 59**	31 at 3 /4 yrs (20–41,58–60)	69*/69*	14 (8/59)//(42 (25/59)
	(**ii**) Nivo 1mg/kg q2w + Ipi		100% IV	**(ii) 56**	64 at 3 /4 yrs (49–76)	36*/36*	4 (2/56)//16 (9/56)
	3 mg/kg q3w x4 vs. (**iii**) Placebo		100% IV	**(iii) 52**	15 at 3 /4 yrs (7–27)	85*/85*	25 (13/52)//46 (24/52)
*Stage II*							
Keynote-716 [10] (NCT03553836)	(**i**) Pem 200 mg q3w **vs.**	III	63% IIB, 35% IIC	**(i) 487**	81 at 2 yr (not reported)	-	-
	(**ii**) Placebo		65% IIB, 35% IIC	**(ii) 489**	73 at 2 yr (not reported)	-	-
CheckMate-76K [11] (NCT04099251)	(**i**) Nivo 480 mg q4w *vs.*	III	60% IIB, 40% IIC	**(i) 526**	89 at 1 year (86–92)	-	-
	(**ii**) Placebo		62% IIB, 38% IIC	**(ii) 264**	79 at 1 year (74–84)	-	-

^a^ The recurrence rate (Rr), estimated for trials with either 3-year or 4-year RFS rates reported, is calculated in the following way: **Rr(%) = 100% − RFSr(%)**; ^b^ patterns of disease recurrence also comprise, apart from local-only recurrence and new distant-only metastasis as reported above in this table, more complex disease recurrence patterns. However, these (more complex) patterns, comprising, e.g., parallel/concomitant local and distant relapse, but also “regional” (or “loco-regional”) relapse, are reported in a non-standardized way across the different trials and publications listed here. Readers are advised to check the cited publications for additional data. RFS rate at 3 years (yrs). Abbreviations: ICI, immune checkpoint inhibitor; Ipi, ipilimumab; Nivo, nivolumab; Pem, pembrolizumab; RFS, recurrence-free survival. Asterisk (*): the 3yr/4yr recurrence rate of the IMMUNED-study [15]. The spacer “(//)” hints at the systematic order of the following data lines: local vs. distant, meaning “local” can be found left from “//” and “distant” on the right hand side.

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
