# Peer review of "Medical Needs and Therapeutic Options for Melanoma Patients Resistant to Anti-PD-1-Directed Immune Checkpoint Inhibition"

_cancers, 2023, doi:10.3390/cancers15133448_

Round 1
Reviewer 1 Report
1. Consider removing author Sickman from the paper. This author is from BMS and represents potential conflict of interest. His contribution can be (and should be) stated at the end of the paper.
2. What is the effect of a patient not completing therapy on "resistance". Is there data on that?
No major concerns
Author Response
Dear Reviewers,
thank you very much for performing the review process for our manuscript. We revised the manuscript according to you suggestions. A detailed point-by-point-reply is given below and a revised manuscript with changes highlighted in yellow is attached.
Please do not hesitate to contact me in case I can be of further assistance.
Best regards
Ralf Gutzmer
Reviewer 1
- Consider removing author Sickman from the paper. This author is from BMS and represents potential conflict of interest. His contribution can be (and should be) stated at the end of the paper.
- What is the effect of a patient not completing therapy on "resistance". Is there data on that?
Reply
- T. Sickmann is indeed an employee of BMS. He was included as coauthor since he not only established the platform for this project by bringing experts from different specialties together. He also provided important intellectual input in all stages of the development of this manuscript. Therefore, his authorship is well justified.
- There are data on patients completing therapy without resistance, e.g., due to a deep partial or complete response or due to study protocols (of Checkmate and Keynote studies) after the completion of the protocol-defined treatment interval. This would be indeed an interesting topic for another article, whereas here we focus on resistance.
Reviewer 2 Report
To the authors,
We applaud this excellent review, and also the intentions of this manuscript, as it highlights what continues to be perhaps the most important challenge in the current treatment of advanced melanoma, that being resistance to anti-PD-1 directed therapy. We submit a few comments/recommendations that may improve the overall impact of the paper.
Introduction:
-first sentence, first paragraph: consider revising grammatically: rather than saying “and tending to”, consider “with a tendency for”.
-last sentence, first paragraph: regarding relatlimab plus nivolumab, please clarify that there is “an advantage of the ICI combination therapy, compared to single agent nivolumab alone”.
-second paragraph, general comment: since adjuvant PD-1 combined with CTLA-4 is not yet a standard approach in the adjuvant setting (outside of clinical trials), I would remove this from the introduction. It might make more sense to briefly mention adjuvant CTLA-4, since that is an available therapy (despite not being used much anymore…)
-second paragraph, general comment: would state with greater emphasis that while adjuvant studies show RFS benefit, they have not yet shown OS benefit.
-first sentence, second paragraph: consider stating that “efficacy of ICI in the unresectable, metastatic setting has led to is testing and approval in the adjuvant setting, etc” since one could argue that resected stage III/IV is also “advanced”.
Materials and methods:
no major comments
Definitions and patterns of response, resistance:
-last sentence, first paragraph: minor grammatical issue: replace “in solid tumors in about 6%”, to say “in solid tumors to be about 6%” .
-last sentence, first paragraph: minor grammatical issue: replace “after ICI, and up to date”, to say “after ICI, but up to date”
Potential mechanisms of resistance
-second paragraph, general comment: to make this section more clear to the audience, would separate the 2nd paragraph into 2 distinct paragraphs, the first to discuss “intrinsic mechanisms”, and the second to discuss “extrinsic mechanisms”.
-Figure 3: under “intrinsic mechanisms”, regarding beta-catenin upregulation, this should say “-> T-cell exclusion”. Not occlusion.
Resistance in the metastatic setting
-general comment: consider more clearly dividing into 1st line and 2nd line studies
-first paragraph, second sentence: please provide reference for acquired resistance rate formula
-second paragraph, general comment: since the discussion of biomarkers applies to all disease settings, consider placing this closer to the end of the manuscript
-third paragraph, general comment: consider moving this earlier in the section, and consider clarifying first that there are overall 3 approaches to first-line PD-1 based therapy: a) PD_1 alone, b) PD-1 with CTLA-4 and c) PD-1 with LAG-3. After this, would then review the other first line combinations.
-third paragraph, last sentence; would clarify that triplet PD-1/BRAF/MEK studies are only for patients with BRAF V600 patients, and would also include a discussion that these were compared against BRAF-MEK and NOT against PD-1, and also that while 2 randomized phase 3 trials were negative, only 1 (IMSPIRE 150) was a positive study, that nonetheless led to FDA approval.
-fourth paragraph, general comment: would place greater emphasis on this section, and try to break down either by larger phase III studies, and or specific mechanisms of interest.
-fourth paragraph, last sentence: keep in mind that the LEAP-003 phase 3 data was recently announced and this did not meet primary OS endpoint
-fifth paragraph, second sentence: citation should be reference 72, not Table 2.
-fifth paragraph, third sentence: there is a phase II, open label study of ipi/nivo that should be discussed here (SWOG S1616, lead author Vanderwalde, NCT03033576).
-sixth paragraph, general comment: the referenced TIL v IPI phase III study here by Rohaan and colleagues should be included in Table 2. Would also consider including discussion of phase II data for Lifileucel, since this is even being filed with FDA.
-seventh and eighth paragraphs, general comment: re: local ablation and oligometastatic progression, consider placing these in a separate section, closer to end of manuscript. Agree completely that these are important clinically, although, it would be better to not mix these in with discussion of systemic approaches to overcome resistance.
-general comment: there is not much discussion of BRAF-MEK inhibition used to overcome PD-1 resistance as a salvage therapy. This should at least be commented on, including perhaps citation of the DREAMseq (EA6134) study.
Brain Metastases
-first paragraph, last sentence: would briefly mention the impact of steroid use on the efficacy of immune checkpoint blockade here, in the context of symptomatic lesions
-third paragraph, general comment: again would at least discuss that BRAF-MEK inhibition has some efficacy in brain mets (COMBI-MB) and could be considered for PD-1 immune failures. There are also triplet studies of PD-1-BRAF-MEK (TRICOTEL and SWOG S200).
-consider discussion of anti-VEGF approaches in MBM
Resistance in the Adjuvant Setting
-second paragraph, second sentence: Delete “The”, which is first word in the sentence. Minor grammatical change.
-third paragraph, general comment: consider discussing that given the different patterns of recurrence, that this may relate to differing mechanisms of resistance (and thus provide rationale for different treatment approaches even)
-last paragraph, last sentence: would note here that more studies are now allowing patients who have had prior adjuvant PD-1 therapy.
Lessons from neoadjuvant studies
-general comment: may want to note that neoadjuvant therapy may allow escalation-descalation of therapy based on response; i.e. those who have resistant disease may need escalated therapy after surgery, and vice versa.
PD-1/PD-L1 Resistant Melanoma: Limitations Perspectives Conclusions
-no comments.
Would review carefully, as there are some small minor typos, but also some sentences that are worded a bit awkwardly.
Author Response
Dear Reviewers,
thank you very much for performing the review process for our manuscript. We revised the manuscript according to you suggestions. A detailed point-by-point-reply is given below and a revised manuscript with changes highlighted in yellow is attached.
Please do not hesitate to contact me in case I can be of further assistance.
Best regards
Ralf Gutzmer
Reviewer 2
To the authors,
We applaud this excellent review, and also the intentions of this manuscript, as it highlights what continues to be perhaps the most important challenge in the current treatment of advanced melanoma, that being resistance to anti-PD-1 directed therapy. We submit a few comments/recommendations that may improve the overall impact of the paper.
Thank you very much for the overall positive assessment and the suggestions for improvement, which were incorporated as detailed below.
Introduction:
-first sentence, first paragraph: consider revising grammatically: rather than saying “and tending to”, consider “with a tendency for”.
Changed as suggested.
-last sentence, first paragraph: regarding relatlimab plus nivolumab, please clarify that there is “an advantage of the ICI combination therapy, compared to single agent nivolumab alone”.
Changed as suggested.
-second paragraph, general comment: since adjuvant PD-1 combined with CTLA-4 is not yet a standard approach in the adjuvant setting (outside of clinical trials), I would remove this from the introduction. It might make more sense to briefly mention adjuvant CTLA-4, since that is an available therapy (despite not being used much anymore…)
A sentence to clarify the approval status was added.
-second paragraph, general comment: would state with greater emphasis that while adjuvant studies show RFS benefit, they have not yet shown OS benefit.
This information is included in the paragraph.
-first sentence, second paragraph: consider stating that “efficacy of ICI in the unresectable, metastatic setting has led to is testing and approval in the adjuvant setting, etc” since one could argue that resected stage III/IV is also “advanced”.
Changed as suggested.
Materials and methods:
no major comments
Definitions and patterns of response, resistance:
-last sentence, first paragraph: minor grammatical issue: replace “in solid tumors in about 6%”, to say “in solid tumors to be about 6%” .
Changed as suggested.
-last sentence, first paragraph: minor grammatical issue: replace “after ICI, and up to date”, to say “after ICI, but up to date”
Changed as suggested.
Potential mechanisms of resistance
-second paragraph, general comment: to make this section more clear to the audience, would separate the 2nd paragraph into 2 distinct paragraphs, the first to discuss “intrinsic mechanisms”, and the second to discuss “extrinsic mechanisms”.
Changed as suggested.
-Figure 3: under “intrinsic mechanisms”, regarding beta-catenin upregulation, this should say “-> T-cell exclusion”. Not occlusion.
Changed as suggested.
Resistance in the metastatic setting
-general comment: consider more clearly dividing into 1st line and 2nd line studies
First line studies are summarized in Table 1 and the first part of this chapter, second line studies in Table 2 and the last paragraph of this chapter.
-first paragraph, second sentence: please provide reference for acquired resistance rate formula
This formula was developed by us for this review. Therefore, there is no reference. We clarified this in the revised manuscript.
-second paragraph, general comment: since the discussion of biomarkers applies to all disease settings, consider placing this closer to the end of the manuscript
We prefer to keep this information in this place.
-third paragraph, general comment: consider moving this earlier in the section, and consider clarifying first that there are overall 3 approaches to first-line PD-1 based therapy: a) PD_1 alone, b) PD-1 with CTLA-4 and c) PD-1 with LAG-3. After this, would then review the other first line combinations.
In the first paragraph of this chapter, we already comment on PD1 mono resistance rates versus nivo+ipi. For nivo+rela, we hesitate to calculate ARr, since we do not have 4-5 year follow up as compared to the other ICI variants. Thus, the ARr rate would be lower due to the shorter follow up.
-third paragraph, last sentence; would clarify that triplet PD-1/BRAF/MEK studies are only for patients with BRAF V600 patients, and would also include a discussion that these were compared against BRAF-MEK and NOT against PD-1, and also that while 2 randomized phase 3 trials were negative, only 1 (IMSPIRE 150) was a positive study, that nonetheless led to FDA approval.
Amended as suggested.
-fourth paragraph, general comment: would place greater emphasis on this section, and try to break down either by larger phase III studies, and or specific mechanisms of interest.
This paragraph was amended accordingly.
-fourth paragraph, last sentence: keep in mind that the LEAP-003 phase 3 data was recently announced and this did not meet primary OS endpoint
Yes, we noticed the press release but there are no data out yet.
-fifth paragraph, second sentence: citation should be reference 72, not Table 2.
Reference was added.
-fifth paragraph, third sentence: there is a phase II, open label study of ipi/nivo that should be discussed here (SWOG S1616, lead author Vanderwalde, NCT03033576).
Information was added.
-sixth paragraph, general comment: the referenced TIL v IPI phase III study here by Rohaan and colleagues should be included in Table 2. Would also consider including discussion of phase II data for Lifileucel, since this is even being filed with FDA.
The Dutch study was added in Table 2 and the information was added in paragraph 6.
-seventh and eighth paragraphs, general comment: re: local ablation and oligometastatic progression, consider placing these in a separate section, closer to end of manuscript. Agree completely that these are important clinically, although, it would be better to not mix these in with discussion of systemic approaches to overcome resistance.
We completely agree that chapter 5 is long and would benefit from a better structure. Therefore, we decided to work with subheading 5.1 (systemic approaches), 5.2 (local approaches), 5.3 (rechallenge).
-general comment: there is not much discussion of BRAF-MEK inhibition used to overcome PD-1 resistance as a salvage therapy. This should at least be commented on, including perhaps citation of the DREAMseq (EA6134) study.
Information and references were added.
Brain Metastases
-first paragraph, last sentence: would briefly mention the impact of steroid use on the efficacy of immune checkpoint blockade here, in the context of symptomatic lesions
This information was added.
-third paragraph, general comment: again would at least discuss that BRAF-MEK inhibition has some efficacy in brain mets (COMBI-MB) and could be considered for PD-1 immune failures. There are also triplet studies of PD-1-BRAF-MEK (TRICOTEL and SWOG S200).
Information added.
-consider discussion of anti-VEGF approaches in MBM
Information added.
Resistance in the Adjuvant Setting
-second paragraph, second sentence: Delete “The”, which is first word in the sentence. Minor grammatical change.
Changed as suggested.
-third paragraph, general comment: consider discussing that given the different patterns of recurrence, that this may relate to differing mechanisms of resistance (and thus provide rationale for different treatment approaches even)
Information added.
-last paragraph, last sentence: would note here that more studies are now allowing patients who have had prior adjuvant PD-1 therapy.
Information added.
Lessons from neoadjuvant studies
-general comment: may want to note that neoadjuvant therapy may allow escalation-descalation of therapy based on response; i.e. those who have resistant disease may need escalated therapy after surgery, and vice versa.
Information added.
PD-1/PD-L1 Resistant Melanoma: Limitations Perspectives Conclusions
-no comments.
Reviewer 3 Report
The main question addressed by the research is the therapeutic options after anti-PD-1 failure. The topic is original and relevant in the field as it provides guidance regarding what to choose after the failure of standard therapy. Compared with other published materials, the sudy adds to the subject area a detailed and systemic summary of medical needs and treatment options after PD-1 failure. The conclusions consistent with the arguments presented in the study and the references are appropriate. Good figures and well designed tables. No further edits required.
Hassel JC, et al. provided a generally well written, thorough review of resistance to anti-PD-1 ICB and subsequent therapeutic choices. This review article is important as it provides insights into the therapeutic resistance mechanisms as well as guidance on daily clinical practices in different settings. I appreciate discussing the resistances by different scenarios, e.g., neoadjuvant, adjuvant, and palliative.
Good.
Author Response
Dear Reviewers,
thank you very much for performing the review process for our manuscript. We revised the manuscript according to you suggestions. A detailed point-by-point-reply is given below and a revised manuscript with changes highlighted in yellow is attached.
Please do not hesitate to contact me in case I can be of further assistance.
Best regards
Ralf Gutzmer
Reviewer 3
The main question addressed by the research is the therapeutic options after anti-PD-1 failure. The topic is original and relevant in the field as it provides guidance regarding what to choose after the failure of standard therapy. Compared with other published materials, the sudy adds to the subject area a detailed and systemic summary of medical needs and treatment options after PD-1 failure. The conclusions consistent with the arguments presented in the study and the references are appropriate. Good figures and well designed tables. No further edits required.
Hassel JC, et al. provided a generally well written, thorough review of resistance to anti-PD-1 ICB and subsequent therapeutic choices. This review article is important as it provides insights into the therapeutic resistance mechanisms as well as guidance on daily clinical practices in different settings. I appreciate discussing the resistances by different scenarios, e.g., neoadjuvant, adjuvant, and palliative.
Reply: thank you for the positive assessment of our work.